# Low-Resource Finetuning for Hallucination Mitigation in Language Models

## Abstract

Hallucinations in Large Language Models (LLMs) pose a significant challenge to their reliable deployment across domains, arising inherently from their design as statistical models that maximize next-token prediction probability based on training data. While methods such as LettuceDetect, RAG-HAT, and prompting techniques have demonstrated efficacy in hallucination detection and mitigation within Retrieval-Augmented Generation (RAG) frameworks, limitations persist. To address these, we propose a novel low-resource hallucination mitigation pipeline that fine-tunes LLMs on synthetic dataset using feedback from LettuceDetect. Our approach reduces hallucination rates in open-source small language models, as validated through evaluations on `RAGTruth` and `PILE-10K` benchmarks. We further discuss the pipeline's extensibility to domain-specific applications.

## 1 Introduction

Hallucinations in Large Language Models (LLMs) pose a significant challenge to their deployment across various domains, as the reliability of generated text is a critical requirement. Hallucinations arise inherently in LLMs due to their nature as fundamentally statistical models optimized for next-token prediction, trained on extensive text corpora. Furthermore, their inevitability is underscored by the Impossibility Theorem established by Karpowicz (2025). This theorem proves that no inference mechanism for an LLM can simultaneously satisfy all four of the following essential properties:

- Truthful (non-hallucinatory) generation.
- Semantic information conservation (i.e., faithful preservation of encoded knowledge).
- Relevant knowledge revelation (exposing facts the model recognizes as useful to the query).
- Knowledge-constrained optimality (producing responses optimized within known facts).

The theorem establishes that any inference process must violate at least one of these objectives. A key implication is the inherent trade-off: diverse applications may necessitate different balances, e.g. safety-critical systems prioritizing truthfulness over completeness. Consequently, achieving perfect hallucination control is mathematically infeasible under realistic inference constraints. Nevertheless, numerous approaches have been explored to detect and mitigate LLM hallucinations, most notably through:

- Mechanistic Interpretability of LLM Internals
- Hallucination mitigation via prompting and grounding in external knowledge sources
- Hallucination detection and mitigation via finetuning

### 1.1 Mechanistic Interpretability of LLM Internals

These methods analyze internal activations, attention-head dynamics, and functional module contributions within LLMs to predict hallucinations, moving beyond output-level criteria (e.g. token probability or uncertainty scores). See e.g. Sun et al. (2024); Yu et al. (2024); Sriramanan et al. (2024); Chen et al. (2024); Du et al. (2024); Ravi et al. (2024); Azaria & Mitchell (2023); Yu et al. (2024); Lee & Yu (2025).

However, several potential limitations exist. Primarily Dubanowska et al. (August 2025) established that hallucination detection techniques relying on internal representations are prone to spurious correlations and fail to demonstrate out-of-distribution generalizability. These results warrant a fundamental reevaluation of the methodology based on internal representations. Furthermore, approaches analyzing internal activations and module dynamics face challenges including model-specific architectural dependencies Yu et al. (2024), absence of causal ground truth linking activations to hallucinations, representational similarity between hallucinated and truthful outputs, sensitivity to noise and prompt variations Chen et al. (2024), high computational costs hindering real-time deployment Sun et al. (2024); Chen et al. (2024), ambiguity in distinguishing hallucinations from creative generations or fine-grained error types Bang et al. (2025), narrow benchmark datasets limiting robustness Sun et al. (2024), and overfitting risks to specific failure modes Sun et al. (2024); Yu et al. (2024). While providing mechanistic insights, these methods remain experimental, lacking the generality, robustness, and efficiency required for production-scale deployment.

## 1.2 HALLUCINATION MITIGATION VIA PROMPTING AND GROUNDING IN EXTERNAL KNOWLEDGE SOURCES

Prompting-based strategies and external knowledge integration offer alternative hallucination mitigation pathways, though some lack token-level granularity in identifying hallucinated content. Key developments include proactive prompting that embeds domain-specific context to enhance factual accuracy Penkov (2024), structured reasoning chains to reduce hallucination rates Braverman et al. (2024), black box revision frameworks for iterative self-correction Mündler et al. (2023), and adaptive RAG hybrids balancing parametric and retrieved knowledge Ding et al. (2024). Empirical evaluations highlight the efficacy of simpler verification-based techniques Barkley & van der Merwe (2024), while decomposition into atomic facts reveals the reasons behind high hallucination rates and categorizes error types Ravichander et al. (2025). Triangulated multi-agent frameworks demonstrate robust detection via consistency analysis Muhammed et al. (2025).

## 1.3 HALLUCINATION MITIGATION VIA FINE-TUNING

Recent studies explore fine-tuning strategies to enhance factual grounding in LLMs:

- **Faithful Fine-Tuning ($F^2$)**: Hu et al. (2024) proposes decomposing QA objectives and applying layer-wise fine-tuning with explicit loss terms to strengthen factual alignment.

- **Noise-Regularized Training**: Khadangi et al. (2025) introduces adaptive Gaussian noise injected into model layers during training, combined with a hybrid loss function.

- **Expertise-Aware Refusal**: Zhu et al. (2025) trains models to decline answering queries beyond their parametric knowledge. The method employs gradient-based sample selection and adaptive weighting to balance accuracy and refusal rates.

- **Finetuning via Hard Sample-aware Iterative Direct Preference Optimization (HIPO)** Hu et al. (2025) introduce LegalHalBench, a benchmark comprising ∼2,000 legal QA instances annotated with five hallucination categories (e.g. incorrect law names, fabricated statutes). They introduce novel metrics—Non Hallucinated Statute Rate, Statute Relevance Rate, and Legal Claim Truthfulness—and develop a two-stage fine-tuning framework combining supervised behavior cloning with HIPO.

- **Hallucination-Focused Preference Optimization** Tang et al. (2025) construct hallucination-focused preference datasets by pairing erroneous and corrected translations, then employ a contrastive preference optimization (CPO) objective for fine-tuning.

- **Fine-tuning Judge Models** Jiang et al. (2025) introduce Bi'an, featuring a bilingual (English-Chinese) benchmark with ∼22k instances across QA, summarization, data-to-text, and translation tasks. They fine-tune lightweight judge models via supervised and preference-based learning, showing their 14B model surpasses larger baselines and is competitive with leading closed-source systems in detecting hallucinations for retrieval-augmented generation.

- **Fine-tuning via Converting Decoder LM into Encoder Detector** ul Islam et al. (2025) train a multilingual hallucination detector using machine-translated versions of the English

FAVA dataset. Validated against human-annotated data in five languages, their model reveals higher hallucination rates in smaller and more multilingual LLMs during long-form QA. The study further reports no correlation between length-normalized hallucination rates and languages' digital resource levels.

## 2 RELATED WORK

### 2.1 RAG-HAT

RAG-HAT Song et al. (2024) is a training pipeline designed to mitigate hallucinations in retrieval-augmented language models. The framework integrates a LlaMa-based hallucination detector with model fine-tuning via Direct Preference Optimization (DPO). GPT-4 Turbo is employed to revise detected hallucinations and the paired (original, revised) outputs guide DPO to align model responses with factual accuracy. A novel aspect involves training the model to explicitly explain its hallucinated content.

Despite its great potential, several limitations warrant consideration:

- **Operational Dependency**: Correction and explanation stages require frequent GPT-4 API calls, introducing cost and external reliability concerns.

- **Potential Bias Transfer**: Reliance on GPT-4 for judgments and revisions risks propagating inherent biases from its training data.

- **Optimization Efficiency**: DPO's slow convergence may prioritize mimicking GPT-4's response style over targeted hallucination reduction, complicating objective alignment.

### 2.2 LUNA

Luna Belyi et al. (2024) is an efficient encoder model based on DeBERTa-large (440M parameters), fine-tuned for hallucination detection in Retrieval-Augmented Generation (RAG) systems. It achieves superior accuracy compared to GPT-3.5 and commercial evaluation frameworks, while reducing inference costs by 97% and latency by 91%. The model supports span-level hallucination detection and employs a novel context chunking mechanism to process exceptionally long sequences. Demonstrating robust cross-domain generalization, Luna exhibits consistent performance across diverse industry verticals and out-of-domain datasets.

### 2.3 LETTUCEDETECT

LettuceDetect Kovács & Recski (2025), inspired by Luna Belyi et al. (2024), is a transformer-based model designed for hallucination detection in Retrieval-Augmented Generation (RAG) systems, specifically evaluating context-answer pairs. It leverages ModernBERT, selected for its extended context window (8192 tokens), which is critical for processing extensive documents to determine answer support accurately. The model's bidirectional architecture captures token relationships across prompts and outputs, enhancing detection efficacy. Evaluated on the RAGTruth dataset Niu et al. (2023), the `lettucedetect-large-v1` variant achieves an F1 score of 79.22%, outperforming prompt-based methods (GPT-4: 63.4%) and approaches the state-of-the-art `LLAMA-3-8B` (83.9%) Song et al. (2024). Despite inference efficiency, operational challenges persist: instruction-tuned models may regenerate answers with new hallucinations even when guided by LettuceDetect's outputs, particularly in complex tasks like summarization. This necessitates iterative refinement, limiting real-world scalability.

## 3 OUR CONTRIBUTION

We posit that hallucinations arise intrinsically from LLM training dynamics. However, given the absence of a formalized hallucination definition it is a great challenge to generalize internal-analysis methods across distributions. Consequently, reducing their prevalence in real-world applications necessitates domain-specific training or fine-tuning to modulate token-to-token relationship statistics.

This work extends the ideas introduced in RAG-HAT and LettuceDetect frameworks. We propose a lightweight fine-tuning pipeline that leverages domain-specialized hallucination detectors. To demonstrate efficacy, we fine-tune compact open-source models on synthetic data generated by `Gemma-3-4b-it`, evaluating performance on the `RAGTruth` and `PILE-10K` benchmarks. The optimization loss derives from `lettucedetect-large-v1`'s token-level hallucination probability estimates, which penalize hallucinated tokens and reward factually grounded generation.

Our method addresses critical limitations of existing approaches:

- **Architecture Agnosticism**: Not constrained to specific model architectures.
- **Resource Efficiency**: Eliminates storage-intensive activation caching for internal-state detectors.
- **Transparency**: Operates without black-box model dependencies (e.g. GPT-4-Turbo).
- **Computational Tractability**: Avoids prohibitive inference costs from large judge models (e.g. `LLaMA-3-70B-Instruct`).
- **Optimization Focus**: Targets hallucination mitigation exclusively, bypassing DPO's slower convergence and stylistic alignment.

Experimental results indicate significant hallucination reduction even for smaller models, with minimal resource overhead. Extending this pipeline to new domains (e.g. medical or legal) requires: (1) curating a small dataset of prompts and human-annotated hallucinatory responses, and (2) training or adapting a bidirectional BERT-style detector (e.g. fine-tuning `lettucedetect-large-v1`) for domain-specific hallucination identification. See also Hu et al. (2025); Pandit et al. (2025).

## 4 DATASETS

### 4.1 TRAIN SET

We generate a synthetic corpus of ∼180k texts using the open-source model `Gemma-3-4b-it`. Each text corresponds to a distinct noun-attribute pair, generated through a three-stage methodological approach:

- **Noun Generation**: `Gemma-3-4b-it` generates 20k common and proper nouns.
- **Attribute Assignment**: For each noun, the model produces 5-15 closely related attributes.
- **Text Generation**: A single text (1024 or 2048 tokens) is created for each noun-attribute pair.

This structured process addresses the inherent challenge of generating diverse, topic-specific texts with open-source LLMs. While the noun-attribute scope is restrictive, it targets a prevalent failure mode in real-world applications: hallucinated attributes and actions in summarization and question answering.

### 4.2 TEST SET

We adopt the RAGTruth dataset Niu et al. (2023) for evaluation. Specifically, we utilize prompts from the Summarization (`Summary`) task, though our framework extends naturally to the Question Answering (`QA`) task. This selection ensures alignment with `lettucedetect-large-v1`'s training distribution, enabling controlled assessment of fine-tuning efficacy. Additionally, we evaluate model perplexity on PILE-10K Nanda (2022), a subset of the PILE dataset, to verify that fine-tuned models retain their initial linguistic capabilities.

## 5 METHODOLOGY

This section delineates our approach for fine-tuning compact open-source language models using feedback from the `lettucedetect-large-v1` model. Our hallucination mitigation pipeline, outlined in Algorithm 1, operates in a single stage.

---

**Algorithm 1** Hallucination Mitigation Finetuning

---

**Input**: $\mathcal{D}$ - train set, $\mathcal{M}$ - LLM, $\mathcal{L}$ - LettuceDetect model
**Parameter**: N - total number of epochs

1: **for** epoch in[1,N] **do**
2:     **T** - sample texts from $\mathcal{D}$
3:     generate responses $\mathcal{A}$ to texts **T** with model $\mathcal{M}$
4:     $\mathcal{P}$ - probability tokens obtained by applying $\mathcal{L}$ on texts **T** and corresponding answers $\mathcal{A}$
5:     calculate logits **Z** of model $\mathcal{M}$ on answers $\mathcal{A}$
6:     calculate loss from $\mathcal{P}$ and logits **Z**
7:     update parameters of model $\mathcal{M}$
8: **end for**

---

## 5.1 Loss

Given the answer $\mathcal{A}$ generated by the LLM to the text prompt **T** the `lettucedetect-large-v1` model outputs hallucination probabilities per token $\mathcal{P} = (p_0, ..., p_s)$, where $s$ is the token sequence length of the answer. Given the logits $\mathbf{Z} = (z_0, ..., z_s)$ calculated by the LLM on the hallucinatory answer $\mathcal{A}$, the loss is calculated as follows:

$$\mathcal{J} = c_l \cdot \sum_k ||l_k||_2^2 + \sum_j z_j \cdot JReLU(p_j, \tau, R) \tag{1}$$

where $\sum_k ||l_k||_2^2$ is the sum of all of the LoRA squared $L^2$ layers norms and $c_l$ is the weight preventing LoRA adapters from overfitting. Following Rajamanoharan et al. (2024), we define Jump ReLU function as follows

$$JReLU(p_j, \tau, R) := \begin{cases} p_j & \text{if } p_j \geq \tau \\ -R & \text{if } p_j < \tau \end{cases} \tag{2}$$

In applications we choose the probability threshold $\tau$ for hallucinatory tokens to be equal to 0.5 and the reward for non-hallucinatory tokens $R$ to be equal to e.g. 1e-4. The choice of $\tau$ is natural due to the choice of hallucination detection model `lettucedetect-large-v1`, i.e. we classify hallucinatory tokens as those for which the output probability from LettuceDetect model is above 0.5. The choice of reward $R$ for non-hallucinatory tokens is crucial as it is a key hyperparameter deciding on whether the model will be rewarded for generating correct text. Setting $R \leq 0$ risks model degradation (overly concise outputs) as was also observed in Song et al. (2024). Furthermore, the reward function $R$ must scale inversely with $B = N \cdot |\mathcal{D}|$, where $N$ denotes the number of training epochs and $|\mathcal{D}|$ represents the cardinality of the training dataset, to mitigate model degradation. Hyperparameter tuning with excessively large values of both $R$ and $B$ risks compromising the model's foundational linguistic capabilities. We emphasize that non-hallucinatory responses incur zero loss to preserve linguistic capabilities.

## 5.2 Finetuning strategy and Parameters

We conducted experiments on H100 80GB GPU card with Python packages **transformers** v.4.53.2, **torch** v.2.7.1 and a fixed seed 43. We evaluated three fine-tuning strategies. In all of them we have choosen AdamW optimizer, $\tau = 0.5$ and $R \in \{1e-5, 1e-4, 5e-4, 1e-3\}$, number of epochs $N$: 1-3. Each hyperparameter configuration was subjected to a single training run within the experimental framework.

- **Full finetuning**: Learning rate: 1e-7 - 1e-8. Batch size: 1-2.

- **LoRA finetuning**: Learning rate: 1e-5 - 1e-6. Batch size: 1-4. $\alpha$: 2-32, $r$: 4-8, $c_l$: 0.1.

- **8-bit Quantized LoRA**: Identical hyperparameters to LoRA.

# 6 LettuceDetect as an Integrated Evaluator and Weak Supervisor

This section outlines the rationale for employing LettuceDetect as a dual-purpose tool—serving as both an imperfect evaluator and a weak supervisor within a unified pipeline. The approach aligns with established practices in weakly supervised learning and addresses inherent limitations through systematic safeguards.

## 6.1 Imperfect Evaluator

The `lettucedetect-large-v1` detector achieves a precision of 64% and recall of 55.8% on `RAGTruth` dataset, significantly exceeding random chance performance, particularly given class imbalance (for Summarization task on RAGTruth ∼30% of the samples are hallucinatory). A random flip (binary classification by chance) would then yield approximately 30% precision and 50% recall, while a naive baseline detector labeling all examples as hallucinatory would achieve ∼30% precision and 100% recall. This indicates that the LettuceDetect captures meaningful data structure. Despite inherent noise, its outputs correlate with ground truth and serve as an informative proxy for highlighting relative differences in model performance.

Evaluating all models with the same detector ensures systematic noise, enabling fair comparisons across experiments. This approach mirrors established surrogate metrics in machine learning (e.g., BLEU, ROUGE, Inception Score), which imperfectly correlate with human judgment yet provide consistent benchmarks for model comparison. Here, the detector is solely used to compare pre- and post-finetuned versions of the same model, not distinct model architectures.

The detector's low recall suggests potential underestimation of true positives, while moderate precision indicates a non-negligible false positive rate. These limitations necessitate cautious interpretation of results.

Human validation remains viable for targeted spot-checks or final evaluations. Integrating this approach mitigates detector limitations, ensuring robust assessment.

## 6.2 Weak supervisor

Training leverages a noisy detector as a weak supervisory signal, substituting for ground-truth annotations. This approach aligns with student-teacher knowledge distillation, where a student model is trained on outputs generated by an imperfect teacher model. Despite the teacher's limitations (64% precision, 55.8% recall, significantly outperforming random baselines on imbalanced data), its outputs encapsulate meaningful data patterns. Consequently, the student model can generalize beyond the teacher's inherent biases, provided the signal retains structural utility.

## 6.3 When is it justifiable?

Employing the same detector for both training and evaluation introduces risks of overfitting to evaluator-specific artifacts. Models may optimize towards "gaming" the evaluator (e.g., exploiting systematic gaps in detecting subtle positives) rather than achieving genuine improvement. This circularity risk necessitates caution, as performance gains may reflect evaluator alignment rather than enhanced real-world capability.

This methodology is defensible when consistency in evaluation standards is prioritized, even with imperfect metrics. To mitigate overfitting, periodic validation via human assessment or alternative metrics is essential. This ensures models do not merely adapt to evaluator idiosyncrasies but achieve robust progress.

## 6.4 Application in the Current Study

The following framework was applied within the current study:

- **Consistency Framework**: Token-level penalties are applied to true and false positives while excluding false negatives. Non-hallucinatory tokens in hallucinated responses are rewarded to prevent model degradation or adversarial optimization.

- **Human verification**: Manual assessment of the LettuceDetect labels on the RAGTruth test split includes systematic checks of true/false positives and random sampling of negatives. Samples transitioning from true positives to false negatives post-finetuning were scrutinized; a high incidence of such transitions indicates adversarial adaptation to systematic gaps in the evaluator, rather than substantive capability enhancement. Conversely, a significant percentage of true negatives—instances correctly classified as non-hallucinations prior to finetuning turning into true hallucinations post-finetuning—indicates model degradation, manifesting as the generation of previously absent hallucinations.

- **Language capabilities preservation** Quantitative assessment ensures foundational language capabilities remain intact post-finetuning, e.g. evaluation on `PILE-10K` dataset Nanda (2022), Gao et al. (2020).

While this approach provides a consistent benchmarking framework, results must be interpreted with acknowledgment of its limitations. Complementary validation safeguards against conflating evaluator-specific gains with substantive progress.

# 7 EXPERIMENTAL RESULTS

This study evaluated five distinct Small Language Models (SLMs) selected for their suitability in on-device deployment scenarios. The models assessed were as follows:

- `LLaMA-3-8B-Instruct` AI (2024); AI@Meta (2024), exclusively quantized to 8-bit precision.

- `Qwen3-1.7B` Alibaba Cloud / QwenLM (2025) , with the "thinking" functionality deactivated.

- `DeepSeek-R1-Distill-Qwen-1.5B` DeepSeek-AI (2025); deepseek-ai (2025), modified to exclude reasoning outputs, retaining only final summaries post-reasoning blocks.

- `Gemma-3-1b-it` Google (2025a); Gemma (2025)

- `Gemma-3-4b-it` Google (2025b)

All models utilized a system prompt enforcing English-only summarization and adherence to language-specific chat templates. Initial observations indicated that `Qwen3-1.7B` and `DeepSeek-R1-Distill-Qwen-1.5B`, whether quantized or operated without the specified prompt, exhibited inconsistent adherence to language constraints, frequently generating outputs in mixed languages rather than producing coherent English summaries.

## 7.1 EVALUATION

This section presents a comparative analysis between base models and their finetuned counterparts, as detailed in Table 1. Each base model underwent distinct finetuning strategies to achieve optimal performance. Comprehensive results for individual strategies are provided in Table 2. Notably, `LLaMA-3-8B-Instruct` was exclusively finetuned in its quantized form.

Two evaluation metrics were employed:

- **Perplexity** measured on the `PILE 10K` dataset Nanda (2022), Gao et al. (2020), with a maximum input sequence length of 1024 tokens and a stride of 512 tokens between consecutive sequences.

- **Non-Hallucinatory Rate (NHR)**, quantified as the percentage of non-hallucinatory responses to summarization prompts from the `RAGTruth` dataset. Model outputs were evaluated using `lettucedetect-large-v1`, classifying responses as hallucinatory if any token was flagged, or non-hallucinatory if all tokens were deemed valid. The responses were generated with hyperparameter values $temperature : 0.7$ and $top_p : 0.7$, $do\_sample : True$, $max\_new\_tokens : 128$.

While non-hallucinatory rate indicates hallucination mitigation efficacy, low perplexity on `PILE 10K` ensures preserved general language capabilities post-finetuning.

The optimal performance, both in Perplexity and Non-Hallucinatory Rate, was achieved with the hyperparameters configuration $R = 1e-5$, $N = 3$. For the full-finetuning strategy a learning rate of $1e-8$, and a batch size of 1. In contrast, the optimal LoRA fine-tuning approach utilized a learning rate of $1e-5$, $alpha = 32$, $r = 8$ in all of the models.

Our approach significantly enhances the Non-Hallucinatory Rate for models exhibiting low Perplexity, specifically `LLaMA-3-8B-Instruct` and `Qwen3-1.7B`, achieving performance levels comparable to highly refined models such as `Gemma-3-1b-it`, `Gemma-3-4b-it`.
However, models `Gemma-3-1b-it`, `Gemma-3-4b-it` demonstrated elevated Perplexity on `PILE 10K` alongside robust baseline performance on `RAGTruth`, exhibiting minor improvement post-fine-tuning. This suggests successful prior instruction-finetuning, while also indicating potential limitations in the judge model or dataset-specific artifacts, with `RAGTruth` serving as the training dataset for `lettucedetect-large-v1`.

Our findings indicate that LoRA finetuning applied to quantized models did not yield improvements in mitigating model hallucinations. Furthermore, we observed that full finetuning of the models `Qwen3-1.7B` and `DeepSeek-R1-Distill-Qwen-1.5B` failed to enhance their performance and, in certain cases, resulted in degradation of capabilities. This suggests that these models exhibit heightened sensitivity to full finetuning procedures.

We emphasize that models were deliberately finetuned on synthetic data to avoid overfitting and spurious correlations inherent in `RAGTruth`'s topical repetitions. This approach prioritizes generalization by decoupling training from dataset-specific features.

Human verification was conducted for each model to ensure that pre-finetuning positives—answers initially flagged as hallucinatory by LettuceDetect—did not transition to false negatives post-finetuning (i.e., true hallucinations erroneously labeled as non-hallucinatory). The emergence of such instances would indicate adversarial adaptation to the evaluator's systematic gaps, rather than substantive model improvement. Additionally, verification confirmed that fewer than 1-2% of pre-finetuning true negatives transitioned to positives post-finetuning. A high percentage of those would suggest model degradation, wherein fine-tuned models generate hallucinations absent in pre-finetuning outputs.

| Model | PERPLEXITY | | NHR | |
| --- | --- | --- | --- | --- |
| | Base | Finetuned | Base | Finetuned |
| LLaMA-3-8B-Instruct-8-bit-Quantized | 8.1992 | 9.1636 | 0.7285 | 0.8228 |
| Qwen3-1.7B | 10.6957 | 11.2493 | 0.7971 | 0.8657 |
| DeepSeek-R1-Distill-Qwen-1.5B | 21.9116 | 22.3263 | 0.6050 | 0.6513 |
| Gemma-3-1b-it | 22.0388 | 22.5979 | 0.7898 | 0.8044 |
| Gemma-3-4b-it | 17.0373 | 21.2813 | 0.8236 | 0.8316 |

Table 1: Evaluation results - base models vs best finetuned model. Perplexity was calculated on `PILE 10K` dataset. Non-Hallucinatory Rate was calculated on prompts from `RAGTruth Summary` task, jointly on both train and test splits.

## 8  CONCLUSIONS AND LIMITATIONS

This work introduces a hallucination-aware finetuning pipeline that extends LettuceDetect and RAG-HAT methodologies, addressing limitations of existing LLM finetuning approaches. The pipeline employs `lettucedetect-large-v1` to identify hallucinations, with detector outputs providing feedback during finetuning. By penalizing incorrect token generation relative to prompt context and rewarding factually grounded generation, the method preserves pretrained language capabilities, accelerates convergence, reduces resource demands, processes large datasets, and avoids dependency on black-box or excessively large models. Our results demonstrate significant hallucination mitigation in small language models without requiring LLM activation storage, prompting techniques, or knowledge graphs. Our approach is compatible with aforementioned complementary methods for

| Model | Split | Base | Full | LoRA | 8-bit QLoRA |
|---|---|---|---|---|---|
| LLaMA-3-8B-Instruct-8-bit-Quantized | TRAIN | 0.7217 | **0.8158** | - | 0.7356 |
| | TEST | 0.7644 | **0.86** | - | 0.7756 |
| Qwen3-1.7B | TRAIN | 0.7870 | 0.7381 | **0.8560** | - |
| | TEST | 0.8511 | 0.7989 | **0.9172** | - |
| DeepSeek-R1-Distill-Qwen-1.5B | TRAIN | 0.5970 | 0.5926 | **0.6471** | - |
| | TEST | 0.6477 | 0.6522 | **0.6733** | - |
| Gemma-3-1b-it | TRAIN | 0.7845 | 0.7894 | **0.7968** | - |
| | TEST | 0.8177 | 0.8278 | **0.8444** | - |
| Gemma-3-4b-it | TRAIN | 0.8158 | 0.8158 | **0.8197** | 0.7898 |
| | TEST | 0.8730 | 0.8777 | **0.8944** | 0.8444 |

Table 2: Non-Hallucinatory Rate. Results on RAGTruth splits, i.e. TRAIN and TEST, for Base models and best models finetuned with each strategy: Full finetuning, LoRA finetuning and 8-bit Quantized LoRA finetuning.

enhanced efficacy.

The pipeline is adaptable to specialized domains (e.g. medical, legal) contingent on existence of domain-specific hallucination detectors. While `lettucedetect-large-v1` offers a foundation, its current training on RAGTruth data limits generalization. Future work should expand detector training to larger, varied datasets. A primary limitation stems from RAGTruth's specificity; broader datasets would enhance detector robustness and domain transferability.

We underscore that our method is aimed at small language models (SLMs) optimized for on-device deployment, where inference speed constitutes a critical constraint. Existing approaches—such as prompting, knowledge graph integration, internal activation analysis, and response regeneration—impose additional computational overhead, increasing latency and memory usage, hindering real-time applicability in resource-constrained settings.

## 9 REPRODUCIBILITY STATEMENT

To ensure reproducibility, this paper provides comprehensive details across the main text. Training dataset creation processes are outlined in section 4. The methodology section 5 describes main algorithm, training procedures, loss function, software and hardware specifications, and hyperparameters (e.g., batch size, learning rate, optimizer). Novel contributions, such as the proposed new loss function and training pipeline, are detailed in subsection 5.1. Tested model architectures are outlined in section 7, while evaluation procedure, metrics and benchmark datasets used for validation are referenced in subsections 7.1 and 6.4.

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

# A APPENDIX

## A.1 LLM USAGE

The LLM was employed for text refinement and proofreading, including correction of grammatical and spelling errors. Additionally, it facilitated the identification of relevant literature, particularly within the domains of Mechanistic Interpretability and Hallucination mitigation through knowledge grounding techniques.

## A.2 EXAMPLES

Below we present representative examples from the `RAGTruth` test split wherein responses from the pre-trained base model were uniformly hallucinatory and nearly identical, whereas responses generated by the fine-tuned models were all correct and non-hallucinatory.

- True positive turning into true negative
  **Prompt**: Summarize the following news within 137 words: Getting caught napping on the job is never good. Getting caught napping on the job in the cargo hold of a plane takes it to a whole different level. Alaska Airlines Flight 448 was just barely on its way to Los Angeles from Seattle-Tacoma International Airport on Monday afternoon when the pilot reported hearing unusual banging from the cargo hold. "There could be a person in there so we're going to come back around, " he told air traffic control. The banging in the cargo hold did come from a person and he turned out to be a ramp agent from Menzies Aviation, a contractor for Alaska Airlines that handles loading the luggage, the airline said. The man told authorities he had fallen asleep. It appears he was never in any danger. The cargo hold is pressurized and temperature controlled, the airline said. The plane was also only in the air for 14 minutes. The passengers knew something wasn't right, almost as soon as the plane took off. "All of a sudden we heard all this pounding underneath the plane and we thought there was something wrong with the landing gear, " Robert Higgins told CNN affiliate KABC. The pounding grew louder. "At that point, we started hearing yelling, screams for help, very, very faint, " Jamie Davis said. "That's when we notified the flight attendant that there was somebody underneath us." As the banging continued, a federal air marshal sprang into action. "At some point, the marshal kind of made himself known," said Troi Ge. "He started banging back, and he yelled really loud and said, 'We're getting ready to land, hold on to something.' " The emergency landing spooked the folks aboard Flight 448. Affiliate KOMO spoke to Marty Collins, another one of the passengers. "We just took off for L.A. regular and then ... about five minutes into the flight the captain came on and said we were going back and we'd land within five to seven minutes, and we did,"Collins said. "When we landed was when all the trucks and the police and the fire trucks surrounded the plane." "I think it's scary and really unsafe, too," Chelsie Nieto told affiliate KCPQ. "Because what if it's someone who could have been a terrorist?" The ramp agent appeared to be in OK after the ordeal. He was taken to an area hospital as a precaution, the airline said. He passed a drug test and was discharged. The employee started work at 5 a.m. and his shift was scheduled to end at 2:30 p.m., just before the flight departed. "During a pre-departure huddle, the team lead noticed the employee was missing. The team lead called into the cargo hold for the employee and called and texted the employee's cell phone, but did not receive an answer. His co-workers believed he finished his shift and went home," the airline's blog said. Alaska Airlines said it's investigating. The man had been on a four-person team loading baggage onto the flight. All ramp employees have security badges, and undergo full criminal background checks

before being hired, according to the airline. After the delay, the flight with 170 passengers and six crew members on board made it to Los Angeles early Monday evening. CNN's Greg Morrison contributed to this report. output:

**Base model answer:** A ramp agent from Menzies Aviation, a contractor for Alaska Airlines, fell asleep in the cargo hold of a plane during a flight from Seattle-Tacoma International Airport to Los Angeles. The pilot reported unusual banging from the cargo hold, and the flight was diverted back to Seattle. The ramp agent was found sleeping and was taken to a hospital as a precaution. He passed a drug test and was discharged. The incident is under investigation by Alaska Airlines. The passengers were unaware of the situation until the pilot announced that the flight was being diverted back to Seattle. The flight was delayed for about an hour before continuing to Los Angeles. The incident highlights the importance of proper security procedures and employee accountability. It also raises questions about the safety and security of cargo holds on commercial flights. The ramp agent's actions were deemed "unacceptable" by the airline, and he may face disciplinary action. The incident is a reminder that even in the most unlikely of places, safety and security must always be a top priority.

**LettuceDetect output**: ["start": 251, "end": 258, "confidence": 0.5677386522293091, "text": " flight", "start": 262, "end": 279, "confidence": 0.5964633226394653, "text": " diverted back to", "start": 615, "end": 624, "confidence": 0.6059316992759705, "text": " about an", "start": 854, "end": 959, "confidence": 0.9523929953575134, "text": " The ramp agent's actions were deemed "unacceptable" by the airline, and he may face disciplinary action."]

**Finetuned model answer:** A man was caught napping in the cargo hold of an Alaska Airlines flight from Seattle to Los Angeles on Monday. The pilot reported hearing unusual banging from the cargo hold and the plane was turned around and returned to the airport. The banging was found to be a ramp agent from Menzies Aviation, a contractor for Alaska Airlines, who had fallen asleep. The man was not in any danger as the cargo hold is pressurized and temperature controlled. The plane was only in the air for 14 minutes. The passengers were aware of something was not right as soon as the plane took off and heard the banging and yelling. A federal air marshal sprang into action and the plane made an emergency landing. The ramp agent was taken to a hospital as a precaution and passed a drug test. The airline is investigating the incident. The man had been on a four-person team loading baggage onto the flight and all ramp employees have security badges and undergo full criminal background checks before being hired. The flight with 170 passengers and six crew members on board made it to Los Angeles early Monday evening. The incident has raised concerns about safety and security. "I think it's scary and really unsafe, too," said one passenger. "Because what if it's someone who could have been a terrorist?"

**LettuceDetect output**: []

- False positive turning into true negative
  **Prompt**: Summarize the following news within 139 words: Every morning, Sissa Abu Dahou puts on traditional male dress, known as a jalabiya, and heads into the streets of the Egyptian city of Luxor for a day of hard work. But this man of the house is actually a woman. The grandmother of two has dressed as a man for 43 years to avoid oppression and eek out a living in the conservative Muslim state. "Why did you do this mother?" her daughter asked during a recorded interview with Egyptian television network CBC. "Your father died and I was six months pregnant with you." she replied. "None of my siblings helped me. I raised you and sent you to school. Without money I could not have gotten you an education." A widow at just 21, Dahou was forced to fend for herself in Egypt's patriarchal south, where decades ago it was unheard of for women to earn their own living. Even in recent years, women make up barely 24% of Egypt's workforce, according to the World Bank. "It is considered wrong that I dressed as a man but no one can judge. Not you or anyone else. Only God can judge me," Dahou said, "People talked but I said I decided to be a man so I can take care of my small daughter." The breadwinner worked as a brick maker for just 25 piasters, the equivalent of a few cents, until she saved up enough money to buy a shoeshine kit. She found a place for her wooden box, painted with red hearts and

the Egyptian flag, alongside the all-male shoe shiners of her community. "If it wasn't for my mother, I would have been on the streets" Houda, her daughter, told TV host Mona al-Shazly through tears. "Honestly I would have been in the streets. I did not find a home except with my mother. And even today my children rely on her." The years of sun and sand weathered and darkened Dahou's face and left her with a voice so deep and raspy she can easily be mistaken for a man. Her only child, Houda, eventually married and had two children of her own. Through the years the one-time housewife dreamed of owning her own business one day, a small street stand to sell snacks and cigarettes. After her interview on CBC television, the governor of Luxor province offered Dahou a kiosk and a cash advance. The maverick had one requirement- she would only agree to meet with the governor in male attire. "Even if I die, I will not take it off," Dahou said as she pulled on the male jalabiya in front of a TV camera, "When I had to dress in a woman's jalabiya when I went to Cairo I felt suffocated. No, I thank God. I don't want anyone to look at me or look at my daughter." Now the 65-year-old who earned respect as a man has gained praise as a woman. Egypt's President, Abdel Fattah el-Sisi, crowned her as one of Egypt's ideal mothers in an official ceremony last month. "If I was really a man, I would not have done this," Dahou told her daughter, "I would have gotten remarried. I would have left you who knows where. You would have been treated without dignity or left homeless. Thank God I was able to protect you. output:

**Base model answer:** Sissa Abu Dahou, a 65-year-old grandmother, has been dressing as a man for 43 years to avoid oppression and make a living in Egypt's conservative society. A widow at 21, she was forced to fend for herself and her daughter, Houda. Dahou worked as a brick maker and later as a shoeshine, earning just a few cents a day. Despite the challenges, she saved up enough to buy a shoeshine kit and eventually started her own business. After an interview on Egyptian TV, the governor of Luxor province offered her a kiosk and cash advance. Dahou has gained praise as a woman, being crowned one of Egypt's ideal mothers by President Abdel Fattah el-Sisi. She remains proud of her decision to dress as a man, saying it allowed her to protect her daughter and provide for her family.
**LettuceDetect output**: ["start": 439, "end": 475, "confidence": 0.7992850542068481, "text": " eventually started her own business"]

**Finetuned model answer:** Sissa Abu Dahou, a 65-year-old grandmother, has been dressing as a man for 43 years to avoid oppression and make a living in the conservative Muslim state of Egypt. She put on traditional male dress, known as a jalabiya, and heads into the streets of the Egyptian city of Luxor for a day of hard work. Dahou, a widow at 21, was forced to fend for herself in Egypt's patriarchal south, where it was unheard of for women to earn their own living. She worked as a brick maker for just 25 piasters, the equivalent of a few cents, until she saved up enough money to buy a shoeshine kit. She found a place for her wooden box, painted with red hearts and the Egyptian flag, alongside the all-male shoe shiners of her community. Dahou's daughter, Houda, said "If it wasn't for my mother, I would have been on the streets" and "Honestly I would have been in the streets. I did not find a home except with my mother. And even today my children rely on her." The governor of Luxor province offered Dahou a kiosk and a cash advance, and she only agreed to meet with the governor in male attire. "Even if I die, I will not take it off"
**LettuceDetect output**: []

- True positive turning into true negative (python code and comments removal)
  **Prompt**: Summarize the following news within 161 words: Blue Bell ice cream has temporarily shut down one of its manufacturing plants over the discovery of listeria contamination in a serving of ice cream originating from that plant. Public health officials warned consumers Friday not to eat any Blue Bell-branded products made at the company's Broken Arrow, Oklahoma, plant. That includes 3-ounce servings of Blue Bell ice cream from this plant that went to institutions in containers marked with the letters O, P, Q, R, S or T behind the coding date. The warning by the Centers for Disease Control and Prevention does not affect other Blue Bell ice cream, including other 3-ounce servings, not made at the plant. But Blue Bell has recalled other products. The company is shutting down the Broken Arrow facility "out of an abundance of caution" to search for a possible cause of

contamination. It is the third time Blue Bell has taken action in light of a listeria outbreak at a Kansas hospital that served the company's ice cream. Listeria monocytogenes was recently found in a cup of ice cream recovered from the hospital. The cup contaminated with the bacteria was produced at the Broken Arrow plant in April 2014, Blue Bell said. And, according to the CDC, listeria bacteria was found in additional samples of the same product that were recovered from the plant. The bacteria in the hospital sample and the factory sample appeared to match each other genetically, the CDC said. But they did not appear identical to listeria samples taken from patients infected in the Kansas outbreak. In a separate outbreak in Texas, the CDC did find that listeria samples taken from patients who came down with listeriosis between 2010 and 2014 in a hospital that served 3-ounce Blue Bell cups matched the listeria in recovered samples. None of this means the ice cream is the source of either spate of the infections. "Investigation to determine whether these illnesses are related to exposure to Blue Bell products is ongoing," the CDC said. In early March, in light of the Kansas listeria outbreak, Blue Bell recalled a group of products made at a plant in Texas. It later added 3-ounce cup servings to the recall. Five people were infected and three died in the past year in Kansas from listeria that might be linked to Blue Bell Creameries products, according to the CDC. All five of them were hospitalized at the same hospital before developing listeriosis, the CDC said. At least four of them had consumed milkshakes made with Blue Bell ice cream before developing the infection. "We are devastated and know that Blue Bell has to be and can be better than this," Paul Kruse, Blue Bell CEO and president, said in a statement. "Quality and safety have always been our top priorities. We are deeply saddened and concerned for all those who have been affected." The CDC advises that individuals and institutions should check their freezers for the recalled products and throw them away. In a statement on its website, Blue Bell said "this recall in no way includes Blue Bell ice cream half gallons, pints, quarts, 3 gallons or other 3 oz. cups." This has been the first product recall in the 108-year history of Blue Bell Creameries, the company said. Listeriosis is a serious infection caused by eating food contaminated with listeria, and primarily affects the elderly, pregnant women, newborns and people with weakened immune systems, according to the CDC. Symptoms of a listeria infection are fever and muscle aches, sometimes associated with diarrhea or other gastrointestinal symptoms. In the United States, an estimated 1,600 people become seriously ill each year, and approximately 16% of these illnesses result in death. Cervical infections caused by listeriosis in pregnant women may result in stillbirth or spontaneous abortion during the second or third trimesters. CNN's Debra Goldschmidt, Amanda Watts and Jacque Wilson contributed to this report. output:

**Base model answer:** Blue Bell ice cream has temporarily shut down one of its manufacturing plants due to listeria contamination. The company has recalled certain 3-ounce servings of ice cream made at the Broken Arrow, Oklahoma, plant. The recall affects products with the letters O, P, Q, R, S, or T behind the coding date. The Centers for Disease Control and Prevention (CDC) has warned consumers not to eat these products. Blue Bell has shut down the plant to search for the cause of contamination. This is the third time the company has taken action due to a listeria outbreak. The CDC is investigating whether the illnesses are related to exposure to Blue Bell products. Listeriosis is a serious infection that primarily affects the elderly, pregnant women, and people with weakened immune systems. Symptoms include fever, muscle aches, and diarrhea. The CDC advises individuals and institutions to check their freezers for the recalled products and throw them away.]] \\end{code} I used the following code to summarize the text: \\n\\n\{code} \\\nimport nltk\nfrom nltk.tokenize import word_tokenize, sent_tokenize \nfrom nltk.corpus import stopwords \nfrom nltk.stem import WordNetLemmatizer \nfrom collections import"

**LettuceDetect output**: ["start": 1080, "end": 1244, "confidence": 0.9907045960426331, "text": "import nltk\nfrom nltk.tokenize import word_tokenize, sent_tokenize \nfrom nltk.corpus import stopwords \nfrom nltk.stem import WordNetLemmatizer \nfrom collections import"]

**Finetuned model answer:** Blue Bell ice cream has temporarily shut down one of its manufacturing plants due to listeria contamination. The Centers for Disease Control and Prevention (CDC) has warned consumers not to eat contaminated products. Blue

Bell has shut down the plant to search for the cause of contamination. It is the third time Blue Bell has taken action in light of a listeria outbreak. The CDC is investigating whether the illnesses are related to exposure to Blue Bell products. Listeriosis is a serious infection caused by eating food contaminated with listeria, and primarily affects the elderly, pregnant women, newborns and people with weakened immune systems.
**LettuceDetect output**: []

- True positive turning into true negative (hallucinatory links removal)
  **Prompt**: Summarize the following news within 85 words: Washington (CNN) In a broad bipartisan vote, the Senate on Tuesday gave final approval to a Medicare reform bill that includes a permanent solution to the "doc fix," a method the government has used to ensure payments to Medicare providers will keep up with inflation. The bill, which passed 92 to 8, also includes a two-year extension of a popular children's health insurance program. The issue of payments to Medicare providers has been a thorny issue for years. Senate Finance Committee Chairman Orrin Hatch of Utah called passage of the bill a "major, major accomplishment." "Tonight, the Senate is voting to retire the outdated, inefficiency-rewarding, common sense-defying Medicare reimbursement system," said Sen. Ron Wyden of Oregon, the top Democrat on the Finance Committee just before the final vote. The House approved the same bill overwhelmingly more than two weeks ago and President Barack Obama is expected to sign it. Senate passage came just hours before cuts to physicians would have taken place since the last temporary "doc fix" had already expired. Some conservative senators, including Republican presidential candidate Ted Cruz of Texas, balked at the more than $200 billion price of the bill and pushed an amendment to have the costs offset. The bill, "institutionalizes and expands Obamacare policies that harm patients and their doctors while adding roughly half a trillion dollars to our long-term debt within two decades," Cruz said in a statement. "Any deal should be fully paid for and include significant and structural reforms to Medicare." But that amendment was defeated, as were several others from each party that came up for votes. Earlier, House Speaker John Boehner of Ohio warned the Senate not to change the bill. "Unless the Senate passes the House-passed 'doc fix,' significant cuts to physicians' payments will begin tomorrow," Boehner said. "We urge the Senate to approve the House-passed bill without delay." Cruz voted against the bill, as did Sen. Marco Rubio of Florida, another Republican running for president. GOP presidential contender Sen. Rand Paul of Kentucky voted for the bill. output:

  **Base model answer:** The Senate has passed a Medicare reform bill that includes a permanent solution to the "doc fix" and a two-year extension of a popular children's health insurance program. The bill passed 92-8 and is expected to be signed by President Obama. The bill addresses a long-standing issue of payments to Medicare providers and includes a cost of over $200 billion. Some conservative senators, including Ted Cruz, opposed the bill, citing concerns about the cost and the expansion of Obamacare policies. The bill is seen as a major accomplishment by Senate Finance Committee Chairman Orrin Hatch.(https://www.cnn.com/2015/03/31/politics/senate-medicare-reform-bill/index.html)]](https://www.cnn.com/2015/03/31/politics/senate-medicare-reform-bill/index.html)]](https://www.cnn.com/2015/03/31/politics/senate-medicare-reform-bill/index.html)]](https://www.cnn.com/2015/03/31/politics/senate-medicare-reform-bill/index.html)]](https://www.cnn.com/2015/03/31/politics/senate-medicare-reform-bill/index",
  **LettuceDetect output**: ["start": 593, "end": 608, "confidence": 0.8013504147529602, "text": "https://www.cnn", "start": 609, "end": 633, "confidence": 0.7843217849731445, "text": "com/2015/03/31/politics/"]

  **Finetuned model answer:** The Senate has given final approval to a Medicare reform bill, which includes a permanent solution to the "doc fix" and a two-year extension of a popular children's health insurance program. The bill passed 92 to 8 and includes a permanent solution to the issue of payments to Medicare providers, which has been a thorny issue for years. The House has already approved the same bill and President Barack Obama is expected to sign it. The bill is a "major, major accomplishment" and will ensure that

payments to Medicare providers will keep up with inflation. Some conservative senators, including Ted Cruz, opposed the bill, citing the cost of over $200 billion and the addition of half a trillion dollars to the long-term debt. However, the bill was passed and is expected to be signed by the President.

**LettuceDetect output**: []

