# OpenReview forum: "Low-Resource Finetuning for Hallucination Mitigation in Language Models"
_ICLR.cc/2026/Conference — Submitted to ICLR 2026_

### Official Review · Reviewer_rLs6 · 2025-10-29

**Soundness:** 2
**Presentation:** 3
**Contribution:** 2
**Rating:** 4
**Confidence:** 4

**Summary:**

In this work, the authors propose a fine-tuning method aimed at reducing hallucinations in small language models. They create a synthetic fine-tuning dataset using a three-step approach: (1) generating 20K nouns with Gemma-3-4b-it, (2) generating 5–15 related attributes for each noun, and (3) producing a text for each noun–attribute pair, resulting in approximately 180K samples. Several small language models are then fine-tuned on this dataset.

During training, a hallucination detector (lettucedetect-large-v1) is used to classify each output token as either a hallucination or non-hallucination. Tokens identified as hallucinations are penalized, while non-hallucinated tokens are rewarded. The models, both before and after fine-tuning, are evaluated on the PILE-10K and RAGTruth datasets to measure perplexity and non-hallucination rate, respectively.

The evaluation results show that while the fine-tuned models exhibit slightly worse perplexity, they achieve a higher non-hallucination rate—indicating reduced hallucination compared to the base models. The authors also find that LoRA fine-tuning outperforms both QLoRA and full fine-tuning.

**Strengths:**

•	The paper tackles an important and highly relevant problem: mitigating hallucinations in decoder-only language models.

•	The proposed fine-tuning approach demonstrates consistent improvements in reducing hallucinations across all evaluated models.

•	The authors include a diverse set of small language models from different families and sizes, which strengthens the generalizability of the findings.

•	The evaluation considers two complementary metrics—perplexity and non-hallucination rate—providing a balanced view of model quality and factual reliability.

**Weaknesses:**

•	While the paper reports hallucination rates, it does not include any task-specific evaluations. Including such experiments could help demonstrate whether the reduced hallucination rate also translates to improved or stable downstream task performance.

•	Although Section 1.3 lists several fine-tuning-based mitigation methods, a direct comparison with one or two of these techniques would make the empirical contribution stronger.

•	It seems that some hyperparameters may have been adjusted using the test set, which could lead to optimistic estimates of performance. Clarifying or separating validation and test data would improve the rigor of the evaluation.

•	Because the same hallucination detector is used for both training and evaluation, improvements on this metric may partially reflect training bias. Using an additional, independent detector (even if less powerful) could help validate the robustness of the results.

•	The results show that while hallucination decreases, perplexity increases somewhat (e.g., from 17 to 21 for Gemma-3-4b-it). It might be helpful to discuss this trade-off and potential strategies to balance factual accuracy and fluency.

•	The related work section is quite detailed and informative but could be made more concise. The saved space could be used to strengthen the empirical section—for example, by including comparisons with other mitigation techniques, visualizing sample fine-tuning data, or showing token-level hallucination rates before and after fine-tuning.

**Questions:**

Q1. Clarity on “Low-Resource” Terminology
The title mentions “low-resource,” which can be confusing since the experiments are conducted in English—a high-resource language. Could the authors clarify in what sense the setting is considered “low-resource”? For example, does it refer to limited model size, limited fine-tuning data, or computational constraints rather than language resource availability?

Q2. Comparison with Existing Mitigation Techniques
The paper introduces an interesting fine-tuning-based approach for hallucination mitigation, but it does not include comparisons with existing methods. Could the authors elaborate on this decision? Were existing techniques difficult to reproduce or incompatible with the proposed setup?

Q3. Evaluation on Downstream Tasks
It would be helpful to understand how the fine-tuned models perform on downstream tasks (e.g., summarization or question answering). Have the authors considered evaluating their models on such tasks to assess whether hallucination reduction affects task-specific performance?

Q4. Hyperparameter Optimization Details
The paper does not clearly specify how hyperparameters were tuned. Could the authors clarify which dataset was used for this purpose? From the current description, it seems that hyperparameters may have been selected based on the RAGTruth test set, which could lead to overfitting. If that’s not the case, additional details on the validation procedure would be appreciated.

---

> ### Author Response · Authors · 2025-11-13
>
> We sincerely thank Reviewer rLs6 for their valuable comments and insightful questions. Their feedback will undoubtedly help us enhance the quality of our paper for future submissions.
>
> Answer to Q1:
> We appreciate the reviewer's insightful comment regarding the clarification of terminology. We think that renaming "Low-Resource" to "Lightweight" in the next version of the paper is valid idea, as this term more accurately reflects the minimal fine-tuning data and computational resources required. Additionally, we would like to emphasize that the primary objective of this research is to demonstrate the applicability of our fine-tuning method to Small Language Models.
>
> Answer to Q2:
> Indeed, we plan to include a comparison of our method with current SOTA methods in the next version of this paper. However, such a comparison has not been feasible so far due to computational constraints and limited access to the powerful open-source or closed LLM models on which these SOTA methods rely. Additionally, smaller open-source LLMs have proven, in our experiments, inadequate for performing the simple tasks required to reproduce SOTA results. We greatly value this feedback and will strive to address this limitation in our future work.
>
> Answer to Q3:
> Our method primarily focuses on RAG scenarios, and we have demonstrated its effectiveness in reducing hallucinations within the summarization task, please refer to the Section 4 for datasets description. In the next version of this paper, we plan to present cross-evaluation results on additional downstream tasks. In particular, we will show how hallucination mitigation fine-tuning on the summarization task also contributes to reducing hallucinations in RAG-based question-answering scenarios.
>
> Answer to Q4:
> We sincerely thank the reviewer for raising this important issue.
> To determine the most critical hyperparameters - namely, R (reward for non-hallucinatory tokens), learning rate, batch size, and LoRA parameters, as detailed in Section 5.2, due to computational constraints we conducted a lightweight grid search. (We note that the parameter \tau was held constant across all experiments and evaluations, as it is the default parameter of the LettuceDetect detector.)
> Our validation set, which comprised part of the synthetically generated dataset (please see Section 4), revealed that the non-hallucinatory rate remained consistent across most parameter combinations. Based on these validation results, we selected the optimal parameters for fine-tuning all Small Language Models.
> As emphasized in Section 7.1, we deliberately decoupled the fine-tuning and validation procedures (including hyperparameter selection) from the final assessment on the RAGTruth dataset. This approach was taken to prevent overfitting to RAGTruth and to avoid spurious correlations inherent in its topical repetitions. For further context on the challenges associated with methods trained and validated on RAGTruth and similar open-source datasets, we refer the reviewer to this work https://aclanthology.org/2025.findings-emnlp.952/
> While the hyperparameters were chosen based on the synthetic validation set, the next version of the paper will include final evaluations with complementary metrics (apart from LettuceDetect and human evaluation) on additional open-source datasets to further validate our findings.

---

> > ### Author Response · Authors · 2025-11-13
> >
> > We appreciate Reviewer rLs6 for raising these important points regarding the limitations of our work. We acknowledge the weaknesses and would like to provide further clarification on those:
> >
> > "• Because the same hallucination detector is used for both training and evaluation, improvements on this metric may partially reflect training bias. Using an additional, independent detector (even if less powerful) could help validate the robustness of the results."
> >
> > Answer:
> > We appreciate this insightful comment. For a detailed discussion on additional human evaluation and the use of the hallucination detector for both training and evaluation, please refer to Section 6. Additionally, Section 7 presents the results of our human evaluation.
> > In the next version of the paper, we will include final evaluations using complementary metrics - beyond LettuceDetect and human evaluation - on additional open-source datasets.
> >
> > "• The results show that while hallucination decreases, perplexity increases somewhat (e.g., from 17 to 21 for Gemma-3-4b-it). It might be helpful to discuss this trade-off and potential strategies to balance factual accuracy and fluency."
> >
> > Answer:
> > We acknowledge the importance of this trade-off, which was discussed in Section 7.
> > Given that the PILE-10K dataset was used to compute model perplexity, it is expected that any fine-tuning procedure would lead to an increase in perplexity on this dataset. Our approach confirms that hallucination mitigation fine-tuning does not degrade the model's language capabilities, as evidenced by ensuring that perplexity does not increase significantly post fine-tuning.
> > Additionally, we note that a higher perplexity in the base model may indicate more extensive instruction fine-tuning - for instance, please compare Qwen3-1.7B with Gemma-3-4b-it.
> > In conclusion, higher perplexity on PILE-10K does not necessarily imply degradation of model's language capabilities.
> >
> > We appreciate this opportunity to clarify these points and remain committed to rigorous evaluation in our future work.

---

> > > ### Comment · Reviewer_rLs6 · 2025-11-27
> > >
> > > I thank the authors for their feedback and clarifications. Since the missing aspects will be addressed in a future version of the paper, I am keeping my scores unchanged.

---

### Official Review · Reviewer_y69o · 2025-10-29

**Soundness:** 2
**Presentation:** 1
**Contribution:** 1
**Rating:** 2
**Confidence:** 4

**Summary:**

This paper proposes a low-resource fine-tuning pipeline for hallucination mitigation in large language models. The method uses LettuceDetect, a pre-trained hallucination detection model, as both a weak supervisor and evaluator. The authors generate a synthetic dataset of noun–attribute pairs using Gemma-3-4B, then fine-tune several small open-source models, including LLaMA-3-8B-Instruct, Qwen3-1.7B, and Gemma variants. The training objective penalizes tokens identified as hallucinated by LettuceDetect and rewards non-hallucinatory tokens, using a Jump ReLU-based loss. Evaluation is conducted on RAGTruth (for hallucination rate) and PILE-10K (for perplexity). The results show modest improvements in non-hallucinatory rate (around 5–8%) with minimal changes in perplexity. The authors claim the approach is architecture-agnostic, computationally efficient, and suitable for small models under resource constraints.

**Strengths:**

1. The topic of hallucination mitigation in language models is relevant and practically important, especially for low-resource or on-device applications.
2. The proposed pipeline is simple, computationally light, and architecture-independent, making it applicable to smaller open models without retraining large detectors.
3. The use of a weak supervision setup is a pragmatic approach to avoid heavy human annotation.
4. The goal of integrating hallucination control into fine-tuning rather than prompting or retrieval methods is conceptually reasonable.

**Weaknesses:**

1. The novelty of the work is minimal. The method merely reuses LettuceDetect as a weak labeler, without introducing new learning objectives, data strategies, or theoretical insights.
2. The experimental validation is insufficient. The model is evaluated on only two small datasets, and the metrics are derived from the same LettuceDetect model used for training, introducing circularity.
3. The writing and structure are poor, and unclear narrative flow, which makes it difficult to follow.
4. The paper contains no figures, diagrams, or visual explanations of the proposed pipeline, which severely limits clarity.
5. The synthetic dataset is overly simplistic and unrelated to the evaluation tasks, weakening the relevance of the training setup.
6. There is no human evaluation or comparison to other fine-tuning strategies like F2, HIPO, or preference optimization methods.
7. The mathematical section offers no real theoretical contribution and introduces unnecessary notation for a simple loss function.
8. The use of LettuceDetect as both training signal and evaluator risks overfitting to detector-specific biases rather than improving genuine factual reliability.

**Questions:**

1. Provide independent evaluation using either human annotation or alternative hallucination metrics to avoid circular reasoning.
2. Include visual diagrams of the training pipeline, loss computation, and evaluation flow to improve readability.
3. Add qualitative examples comparing pre- and post-fine-tuning responses to illustrate concrete behavioral changes.
4. Perform ablation studies to separate the effects of LettuceDetect supervision, loss configuration, and fine-tuning strategy.
5. Analyze efficiency in terms of training time, memory usage, and scalability across model sizes.
6. Reorganize and clean the manuscript for clarity, consistent formatting, and concise writing.
7. Compare quantitatively against recent fine-tuning and preference optimization approaches to position the contribution properly.
8. Discuss the limitations of using a noisy weak supervisor and suggest potential methods to mitigate detector bias.
9. Given the small technical novelty and weak empirical results, the paper would be more suitable as a workshop or system report rather than a full ICLR submission.

---

> ### Author Response · Authors · 2025-11-13
>
> We appreciate Reviewer y69o's feedback and having the opportunity to clarify several points that appear to stem from misunderstandings of our submission and which do not reflect the paper's content. Below, we address the specific concerns raised:
>
> "Q1 Provide independent evaluation using either human annotation or alternative hallucination metrics to avoid circular reasoning."
> Answer: The human evaluation methodology and our approach to avoiding circular reasoning are thoroughly discussed in Sections 6 and 7.1.
>
> "Q2 Include visual diagrams of the training pipeline, loss computation, and evaluation flow to improve readability."
> Answer: The training pipeline is visualized in an algorithm diagram, and the loss computation and evaluation metrics are detailed in Sections 5.1 and 7.1. Given the simplicity of these components, we are convinced that additional visual diagrams would not enhance readability.
>
> "Q3 Add qualitative examples comparing pre- and post-fine-tuning responses to illustrate concrete behavioral changes."
> Answer: Illustrative examples of responses pre- and post-fine-tuning are provided in the Appendix.
>
> "Q4 Perform ablation studies to separate the effects of LettuceDetect supervision, loss configuration, and fine-tuning strategy."
> Answer: Full ablation studies will be included in the next version of the paper. Currently, Table 2 presents results of different fine-tuning strategies. As noted in our response to Reviewer rLs6:
> " To determine the most critical hyperparameters - namely, R (reward for non-hallucinatory tokens), learning rate, batch size, and LoRA parameters, as detailed in Section 5.2, due to computational constraints we conducted a lightweight grid search. (We note that the parameter \tau was held constant across all experiments and evaluations, as it is the default parameter of the LettuceDetect detector.) Our validation set, which comprised part of the synthetically generated dataset (please see Section 4), revealed that the non-hallucinatory rate remained consistent across most parameter combinations. "
>
> "Q8 Discuss the limitations of using a noisy weak supervisor and suggest potential methods to mitigate detector bias."
> Answer: Section 6 provides a comprehensive discussion on the use of a weak noisy supervisor and strategies to mitigate detector bias.
>
> We thank the reviewer for raising questions Q5, Q6, and Q7, which will be addressed in the next version of the paper, as partially noted in our response to Reviewer rLs6.

---

> > ### Comment · Reviewer_y69o · 2025-11-24
> > **thanks**
> >
> > Thank you for your response. However, my concerns were not addressed, so I will keep my score unchanged.

---

### Official Review · Reviewer_iefr · 2025-11-01

**Soundness:** 1
**Presentation:** 2
**Contribution:** 2
**Rating:** 2
**Confidence:** 4

**Summary:**

The paper proposes a low-resource fine-tuning pipeline that uses a token-level hallucination detector (LettuceDetect) as both weak supervisor and evaluator. The loss penalizes tokens the detector flags and lightly rewards tokens considered grounded, with LoRA, QLoRA, and full fine-tuning variants. Training uses about 180k synthetic noun-attribute texts and evaluation reports Non-Hallucinatory Rate on RAGTruth plus perplexity on PILE-10K. Reported gains in NHR are clearest for smaller models, while perplexity generally increases after fine-tuning.

**Strengths:**

1. Nice idea and approach - Token-level supervision for hallucinations is a sensible way to train model with dense signals and could be architecture-agnostic and cheap to run.

2. Training dataset is simple - The noun-attribute generator is straightforward and scales, which makes the recipe easy to reproduce.

**Weaknesses:**

1. Flawed evaluation - The same detector is used both for supervision and evaluation. The detector’s own precision and recall on RAGTruth are modest, which raises the risk that the model learns the judge rather than factuality. Human checks are described only at a high level, without any details that would establish reliability. Consider adding additional evaluations such as FAVA-Bench [1] or FactScore [2] to distinguish overfitting from generalization.

2. Lack of baselines - Results compare only base vs fine-tuned models. This problem is widely studied and has established previously proposed alternatives. At minimum include SFT on the same data with standard cross-entropy, refusal tuning [3], “corrected data” training, and simpler token-weighting or DPO variants to test whether the detector-guided loss is essential. Effects of data size are also important to study. The paper itself mentions many prior work on finetuning models for addressing hallucination which should be considered as baselines

3. Flawed perplexity findings - The paper’s narrative suggests mixed perplexity effects, yet Table 1 shows perplexity increases for all listed models after fine-tuning, which implies degradation in general language modeling. Additional evaluations for broader capability checks such as MMLU or GLUE need to be done ensure core skills are not harmed.

References
[1] FAVA-Bench - Mishra, A., Zhou, Y., Wang, S., et al. Fine-grained Hallucination Detection and Editing for Large Language Models. arXiv:2401.06855, 2024.

[2] FActScore - Min, S., Krishna, K., Lyu, X., et al. FActScore: Fine-grained Atomic Evaluation of Factual Precision in Long-Form Text Generation. EMNLP 2023 (Main), 2023. ACL Anthology+1

[3] Refusal tuning - Zhang, H., Diao, S., Lin, Y., et al. R-Tuning: Instructing Large Language Models to Say “I Don’t Know”. NAACL 2024 (Long), 2024.

**Questions:**

see weaknesses.

---

> ### Author Response · Authors · 2025-11-13
>
> We sincerely thank Reviewer iefr for their insightful comments, which will help us improve future versions of this paper. Below, we address the specific points raised:
>
> Weaknesses:
> "Flawed evaluation - The same detector is used both for supervision and evaluation. The detector’s own precision and recall on RAGTruth are modest, which raises the risk that the model learns the judge rather than factuality. Human checks are described only at a high level, without any details that would establish reliability. Consider adding additional evaluations such as FAVA-Bench [1] or FactScore [2] to distinguish overfitting from generalization."
>
> Answer:
> We appreciate this insightful comment regarding our use of the hallucination detector. The discussion of employing the detector for both supervision and evaluation is presented in Section 6. While we recognize the limitations inherent in using an imperfect detector, we have addressed this through human evaluation, as detailed in Section 7.1.
> Our research does not claim that LettuceDetect serves as a definitive solution for hallucination mitigation. Rather, it demonstrates the potential of our proposed method. We anticipate that with a more precise hallucination detector - particularly in specialized domains such as medical or legal - this fine-tuning approach could yield substantial improvements in LLM performance.
> We are grateful for the suggestion to incorporate FAVA-Bench and FactScore evaluations. Indeed, we already plan to expand our assessments by including additional hallucination detectors and datasets in future work.
>
>
>
> "Lack of baselines - Results compare only base vs fine-tuned models. This problem is widely studied and has established previously proposed alternatives. At minimum include SFT on the same data with standard cross-entropy, refusal tuning [3], “corrected data” training, and simpler token-weighting or DPO variants to test whether the detector-guided loss is essential. Effects of data size are also important to study. The paper itself mentions many prior work on finetuning models for addressing hallucination which should be considered as baselines."
>
> Answer:
> As noted in our response to Reviewer rLs6, we will add baseline comparisons in the next version of the paper.
>
>
>
> "Flawed perplexity findings - The paper’s narrative suggests mixed perplexity effects, yet Table 1 shows perplexity increases for all listed models after fine-tuning, which implies degradation in general language modeling. Additional evaluations for broader capability checks such as MMLU or GLUE need to be done ensure core skills are not harmed."
>
> Answer:
> We appreciate this important observation regarding perplexity effects. To clarify, our paper does not suggest any mixed perplexity effects. As anticipated, the fine-tuned model exhibits higher perplexity compared to its base counterpart, which is an expected consequence of the adaptation process and changes in token-to-token relationships rather than an indication of model degradation.
> We will explicitly verify this finding through comprehensive evaluations on MMLU and GLUE benchmarks in the next version of the paper, and we thank the reviewer for this valuable suggestion.
> For additional context, we would like to reference our response to a similar question from Reviewer rLs6:
> "Given that the PILE-10K dataset was used to compute model perplexity, it is expected that any fine-tuning procedure would lead to an increase in perplexity on this dataset. Our approach confirms that hallucination mitigation fine-tuning does not degrade the model's language capabilities, as evidenced by ensuring that perplexity does not increase significantly post fine-tuning. Additionally, we note that a higher perplexity in the base model may indicate more extensive instruction fine-tuning - for instance, please compare Qwen3-1.7B with Gemma-3-4b-it. In conclusion, higher perplexity on PILE-10K does not necessarily imply degradation of model's language capabilities."

---

> > ### Comment · Reviewer_iefr · 2025-11-27
> > **Response to Author comment**
> >
> > Thank you for the response to my concerns. Since the authors say that they plan to address all the empirical concerns (ablations, broader evaluations, etc) in next iteration of the paper, I will maintain my current rating as the paper in the current form is not complete.

---

### Official Review · Reviewer_bA6B · 2025-11-01

**Soundness:** 1
**Presentation:** 2
**Contribution:** 1
**Rating:** 2
**Confidence:** 4

**Summary:**

This paper proposes a low-resource finetuning pipeline designed to mitigate hallucinations in Small Language Models (SLMs), particularly within Retrieval-Augmented Generation (RAG) contexts. Unlike existing methods that often rely on large, expensive "judge" models like GPT-4 (e.g., RAG-HAT), this work employs a lightweight, open-source hallucination detector, LettuceDetect, as a "weak supervisor".

**Strengths:**

The pipeline is resource-efficient and practical. By avoiding reliance on large, closed-source models (like GPT-4) or complex optimization methods (like DPO), it provides an accessible pathway for improving SLM factuality, especially for on-device applications.
This is possible by introducing a loss function (JReLU) specifically tailored to the task of token-level hallucination mitigation. This allows for direct optimization against the detected undesirable behavior, rather than just mimicking a style. (unlike DPO)

The method demonstrates strong empirical results when combined with LoRA. It successfully improves the Non-Hallucinatory Rate (NHR) for several SLMs, showing its potential utility.

**Weaknesses:**

The proposed training pipeline appears to be incomplete and fundamentally flawed. Its success is entirely dependent on the implicit regularization of LoRA. The experiments clearly show that when using Full-Finetuning, the method fails and even degrades model performance (Table 2). This indicates the pipeline, on its own, is unstable.

The evaluation is very narrow. It only reports the target metric (NHR on RAGTruth) and a general language metric (Perplexity on PILE-10K). There is no evaluation on standard downstream benchmarks (e.g., NQ, or standard open-ended QA tasks). This makes it impossible to assess if the finetuning has catastrophically damaged the model's general reasoning and knowledge capabilities, which is a critical concern given the observed Perplexity degradation.

The paper states that the reward hyperparameter R for non-hallucinatory tokens is "crucial", yet it fails to provide a principled methodology for its selection. The optimal choice of R=1e-5 appears to be the result of a simple grid search and not discussed properly.

**Questions:**

* Given that the pipeline fails under full finetuning, does the paper suggest that the core contribution is not the pipeline itself but rather the finding that this JReLU loss function only works when combined with a strong implicit regularizer like LoRA?

* A clear trade-off between NHR improvement and Perplexity degradation is shown. From a practical standpoint, do you believe this degradation in fundamental language capability is an acceptable price to pay for the observed NHR gains?

* Why did you not report performance on standard downstream benchmarks (e.g., open-ended QA tasks)?

* Why was the JReLU threshold $\tau$ arbitrarily fixed at 0.5, rather than being treated as a critical hyperparameter to be calibrated on a validation set? Given that similar thresholds are pivotal for performance in many works (e.g., conformal/selective prediction and uncertainty quantification), this "natural" choice seems unprincipled and suboptimal.

---

> ### Author Response · Authors · 2025-11-13
>
> We appreciate the Reviewer bA6B's comments and engagement with our work. However, we would like to clarify several points that appear to be based on misunderstandings of our paper's content and findings.
>
> Questions:
> Q1. Given that the pipeline fails under full finetuning, does the paper suggest that the core contribution is not the pipeline itself but rather the finding that this JReLU loss function only works when combined with a strong implicit regularizer like LoRA?
>
> Answer:
> The paper does not suggest that the pipeline fails under full finetuning. Our experiments were initially conducted with LoRA, and through grid search over the parameters mentioned in Section 5.2, we selected optimal parameters (e.g., R and number of epochs) based on performance on the synthetic validation set (not RAGTruth). These same parameters were then applied to full finetuning runs. Due to resource constraints, we were only able to perform 1-2 full finetuning runs for each model, and we reported the results from these runs. This limited sampling does not imply that the pipeline fails under full finetuning.
> As noted in Section 7.1, we observed that full finetuning of Qwen3-1.7B and DeepSeek-R1-Distill-Qwen-1.5B did not enhance their performance and, in some cases, led to degradation. This suggests that these smaller models exhibit heightened sensitivity to full finetuning procedures.
> Contrary to the reviewer's claim, the full finetuning procedure worked effectively in its first application to LlaMa-3-8B-Instruct (8-bit quantized), where it outperformed LoRA finetuning. This indicates that the issue is not with the pipeline itself but with the sensitivity of smaller models to full finetuning.
> In the next version of the paper, we will present results for Qwen-3-8B and a family of larger DeepSeek models, demonstrating that full finetuning performs better than LoRA for larger models. This further supports the hypothesis that the superior results with LoRA in smaller models stem from their general sensitivity to full finetuning, which is mathematically intuitive given their compact and highly optimized architectures. For instance quantization of these small models (e.g., Qwen, DeepSeek, Gemma) can lead to language mixing and significant loss of capabilities.
>
> Q2. A clear trade-off between NHR improvement and Perplexity degradation is shown. From a practical standpoint, do you believe this degradation in fundamental language capability is an acceptable price to pay for the observed NHR gains?
>
> Answer:
> We appreciate this important question about the relationship between Non-Hallucination Rate (NHR) and perplexity. As we noted in our response to Reviewer rLs6, this trade-off was already acknowledged:
> "We acknowledge the importance of this trade-off, which was discussed in Section 7. Given that the PILE-10K dataset was used to compute model perplexity, it is expected that any fine-tuning procedure would lead to an increase in perplexity on this dataset. Our approach confirms that hallucination mitigation fine-tuning does not degrade the model's language capabilities, as evidenced by ensuring that perplexity does not increase significantly post fine-tuning. Additionally, we note that a higher perplexity in the base model may indicate more extensive instruction fine-tuning - for instance, please compare Qwen3-1.7B with Gemma-3-4b-it. In conclusion, higher perplexity on PILE-10K does not necessarily imply degradation of model's language capabilities."
> We would like to clarify that the higher perplexity observed in fine-tuned models on the PILE-10K dataset does not indicate model degradation. In the next version of this paper, we will verify through MMLU and GLUE benchmarks that the fine-tuned models maintain their general language capabilities.
> Furthermore, we emphasize that our lightweight hallucination mitigation fine-tuning approach is designed to be compatible with other alignment methods such as DPO, SFT, and distillation. This compatibility allows for the development of less hallucinatory models while preserving their language capabilities.

---

> > ### Author Response · Authors · 2025-11-13
> >
> > Q3. Why did you not report performance on standard downstream benchmarks (e.g., open-ended QA tasks)?
> >
> > Answer:
> > We thank the Reviewer for this comment, additional reports will be added in the next version of the paper.
> >
> > Q4. Why was the JReLU threshold arbitrarily fixed at 0.5, rather than being treated as a critical hyperparameter to be calibrated on a validation set? Given that similar thresholds are pivotal for performance in many works (e.g., conformal/selective prediction and uncertainty quantification), this "natural" choice seems unprincipled and suboptimal.
> >
> > Answer:
> > We appreciate the Reviewer's suggestion as we plan to incorporate extended evaluations in the next version of this paper. While we could conduct additional experiments with different \tau values to demonstrate further improvements through calibration on the synthetic validation set, we opted to use \tau=0.5 - the default threshold value in the LettuceDetect model for marking hallucinatory tokens - due to resource constraints. This choice allowed us to show that meaningful reductions in hallucinations can be achieved even without further calibration.

---

> ### Author Response · Authors · 2025-11-13
>
> Weaknesses:
> "The proposed training pipeline appears to be incomplete and fundamentally flawed. Its success is entirely dependent on the implicit regularization of LoRA. The experiments clearly show that when using Full-Finetuning, the method fails and even degrades model performance (Table 2). This indicates the pipeline, on its own, is unstable."
>
> Answer: We hope that we already dispelled the doubts the Reviewer has by answering the Q1.
>
>
> "The evaluation is very narrow. It only reports the target metric (NHR on RAGTruth) and a general language metric (Perplexity on PILE-10K). There is no evaluation on standard downstream benchmarks (e.g., NQ, or standard open-ended QA tasks). This makes it impossible to assess if the finetuning has catastrophically damaged the model's general reasoning and knowledge capabilities, which is a critical concern given the observed Perplexity degradation."
>
> Answer: We have partially addressed these considerations in our discussion of perplexity in the response to Q2. As noted there, we will expand the evaluation framework in the next version of the paper to provide more comprehensive analysis of these aspects.
>
> "The paper states that the reward hyperparameter R for non-hallucinatory tokens is "crucial", yet it fails to provide a principled methodology for its selection. The optimal choice of R=1e-5 appears to be the result of a simple grid search and not discussed properly."
>
> Answer:
> Due to resource constraints, we conducted a simple grid search to identify an effective reward value (R=1e-5) for LoRA fine-tuning (and full fine-tuning in the case of the LlaMa model). In Section 5.1, we elaborate on the rationale behind selecting this value. As discussed in the paper "Rag-hat: A hallucination-aware tuning pipeline for LLM in retrieval-augmented generation," similar findings were observed - specifically, that failing to reward the model for correct generation can lead to model degradation.

---

> ### Comment · Reviewer_bA6B · 2025-11-25
>
> I thank the authors for their response.
> However, my primary concerns remain unaddressed since the necessary empirical backing is not provided (e.g., other benchmarks, full fine-tuning on larger models).
>
> Therefore, I will maintain my rating.

---

### Meta-Review · Area_Chair_dg8x · 2026-01-06

**Summary:**

This paper proposes a resource-efficient finetuning pipeline to reduce hallucinations in Small Language Models within Retrieval-Augmented Generation contexts. The approach uses a lightweight open-source hallucination detector (LettuceDetect) as a weak supervisor which introduces a JReLU loss function which rewards tokens considered grounded, with LoRA, QLoRA, and full fine-tuning variants.

While the research direction is valuable, the submission requires substantial additional experimentation before publication. The authors' responses largely promise future work rather than addressing concerns with existing evidence.

**Reviewer Concerns:**

**Addressed Concerns**
- Concerns about core contributions, takeaways and clarifications on the method.

**Critical concerns that remain unaddressed:**
- The empirical evaluation scope is insufficient. The paper reports only NHR and perplexity, omitting standard downstream benchmarks that would reveal whether the finetuning damages general capabilities. The authors acknowledge this gap and promise future additions, but such fundamental evaluations should be present in the submitted version.
- The method seems to have a bias: the same hallucination detector is used for both training and evaluation. The authors respond with pointers to human evaluation, but this seems somewhat unsatisfactory.
- Comparison with baselines seems limited: only base model and adapted models are compared, instead of other hallucination mitigation approaches to the proposed approach.
- Finally, the quality of generation, as determined by perplexity, is degraded after introducing the JReLU loss.

**Reviewer Scores:**

All reviewers explicitly acknowledge that they will not be changing their scores based on the response.

---

### Decision · Program_Chairs · 2026-01-26

Reject